# Unveiling the Potential of Machine Learning Applications in Urban Planning Challenges

**Sesil Koutra** [1] and **Christos S. Ioakimidis** [2,*]

1   Faculty of Architecture and Urban Planning, University of Mons, 88 Str. Havré, 7000 Mons, Belgium
2   Inteligg P.C., Karaiskaki 28, 10554 Athens, Greece
*   Correspondence: cioakim@inteligg.com

**Abstract:** In a digitalized era and with the rapid growth of computational skills and advancements, artificial intelligence and Machine Learning uses in various applications are gaining a rising interest from scholars and practitioners. As a fast-growing field of Artificial Intelligence, Machine Artificial Intelligence deals with smart designs, data mining and management for complex problem-solving based on experimental data on urban applications (land use and cover, configurations of the built environment and architectural design, etc.), but with few explorations and relevant studies. In this work, a comprehensive and in-depth review is presented to discuss the future opportunities and constraints in meeting the next planning portfolio against the multiple challenges in urban environments in line with Machine Learning progress. Bringing together the theoretical views with practical analyses of cases and examples, the work unveils the huge potential, but also the potential barriers of the complexity of Machine Learning to urban planning strategies.

**Keywords:** case-study analysis; machine learning; urban planning

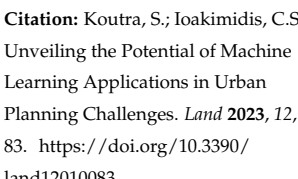



## 1. Introduction

Digitalization is gaining rising interest in all fields of daily life, being favored by the increasing computational capacities and the emergence of efficient algorithmic processes which facilitate data mining. In line with this, Machine Learning (ML) as an intersection of informatics and statistics is a promising challenge for more evidence-based decisions [1] to fill in the gap of existing technological tools and instruments for spatiotemporal requirements. Bhavsar et al. [2] define ML as a collection of data-driven models to automate data through significant patterns, while the first attempts to develop machines to imitate living behavior dates to the 30s by Ross [3]. In 1959, Samuel approaches the concept as the '*field of study that provides computes with the ability to learn without being further programmed*' [4].

As living laboratories in a multidimensional context with tremendous environmental and social challenges, cities are being more and more implicated in these applications, especially those oriented towards meeting the complex ambitions of sustainability, resilience and climate adaptation, to cite some of them, and dealing with a noticeable mass of data ([2,3]). At the same time, rapid urbanization challenges and quality of life (QoL) degradation puts pressure on planners to channel the growth and provide monitoring strategies, while the traditional methods (e.g., surveys, etc.) are time-consuming with insufficient outcomes. Advancements in urban geography and relevant sciences, commonly geographical information systems (GIS) tools, use simulations to evaluate and analyze the complex interactions in a city with limited efficiency to simulate scenarios for future growths [4,5] and visualize spatial, demographic and other relevant data to benefit from digital innovations and patterns. These are the key transformations needed for the abovementioned roadmaps.

Combined with the technology and software advancements, ML and the field of Artificial Intelligence (AI) are being prioritized and becoming more and more essential

for cities' operations towards smart solutions, e.g., optimization of energy performance or waste management, etc. ([6,7]). They are adopted widely for diverse tasks of the digital society while reducing the human effort [8], a recognition of the achievements in data acquisition and the practical use of algorithmic approaches [9]. Many scholars have already stressed the importance of ML for accurate predictions and correlations between spatial indicators (e.g., [10,11]).

As ML transcended the conventional techniques of modeling, a huge potential of big data management to address complex city problems is presented at the crossroads of modern urban planning challenges to make up their dynamics [12,13]. Broadly speaking, ML consist of a group of different models and patterns with the ability to minimize error using repeated processes from data collection, analysis and monitoring [14]. Based on the existing definitions, Machine Learning consists of a set of techniques to automatically detect and predict data or perform decision-making processes under an important level of uncertainty [15]. Hence, ML consists of methods leading to evidence-based processes to meet the standards and quality of a complex problem. Its rapid evolution and growth, with the parallel emergence of its potential, will equal the challenges of modern cities in several fields (mobility, energy, etc.). More and more cities are being included in this dynamic, which concerns the drivers of the urban functionalities or decision-making processes optimizing performance and leading to automation. Overall, the existing ML demonstrations on urban and spatial problems consist of spatiotemporal subjects [16]; however, their implication has not yet been fully explored, despite their large repertoire [17].

Despite the technological achievements and the progress in ML uses, data availability remains a sophisticated task and not equally distributed in every corner of the world. Lack of standards, the topics of private life and confidentiality, spatial granularities or even the lack of synergies and the nature of the heterogeneous data hinder the ML operation. On the other hand, the applications of ML algorithms on specific fields, such as geography, demonstrate the complexity of benchmarking the relevant studies due to the type of data used for the ML analyses or the missing parameters [18].

Hence, the objective of this work is to provide a critical review of the literature on ML methods and urban modelling applications at the crossroads of urban complexities as a promising research area for the years to come, associated with their advantages and opportunities for the favor of efficient data management [19]. At the same time, the aim of the work is to identify the gaps and significant constraints regarding this enterprise and to define future directions with a comprehensive benchmarking analysis and in-depth reviews to evaluate the future challenges. We also discuss the potentials and constraints of ML with a portfolio of analyzed examples to identify the future directions for ML applications sufficient to meet next generation planning strategies. Overall, the work addresses the unexplored gaps in ML's role in urban analysis in a scoping review and unveils the most prominent approaches.

The remainder of the paper is organized as follows (Figure 1). Section 2 broadly describes the taxonomy of the contents of the review for urban applications related to ML uses. In Section 3, different challenges are discussed with respect to the built environment, land uses and the transportation, to cite the most important among them. Section 4 provides a portfolio of concrete examples and cases to retrieve lessons learnt, and Section 5 concludes the review.

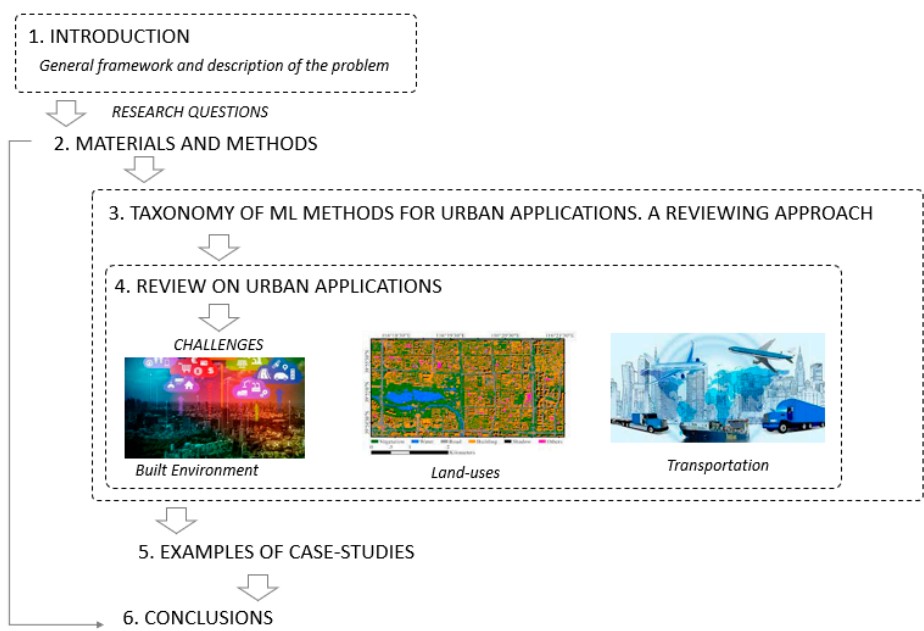

**Figure 1.** Paperwork plan and structure.

## 2. Materials and Methods

Methodologically, the chart of the study is organized in the conventional flow of a scoping review including the planning criteria for the selection of the relevant documentation, the identification and screening of the relevant scientific sources and the analysis and crossing of findings.

As keywords core to the study, ML and urban applications remain vast and inadequate for the consolidation of accurate outcomes. To identify an initial branch of sources, the Scopus platform was used to include a broad, transversal coverage to guarantee reproducible and accurate results using the bibliometric approach provided by the VOS viewer tool. More than 2000 scientific sources were detected in the Scopus platform.

For this scope, the work focuses on:

- The investigation of the spatial and temporal distribution of the selected sources with necessary filtering of the key information of title, authors, years and keywords and focusing mainly on English-language publications;
- The identification of sources by type of data provided (e.g., open or not);
- The chronological constraint from 2000 onwards;
- The identification of specific keywords and research areas (computer science, engineering, environmental and among others) (Figure 2).

While spatial data mining has been accelerated through technological advancements, the availability is not equally allocated throughout the world (Figure 5) ([20,21]). Complementary to this, Casali et al. [18] spatialized 159 related scientific documents distributed in different countries of the world and over time confirmed the discrepancies between them (Figure 3); most of the cases appeared in China and US, followed by the UK and overall only 31% were detected in Europe.

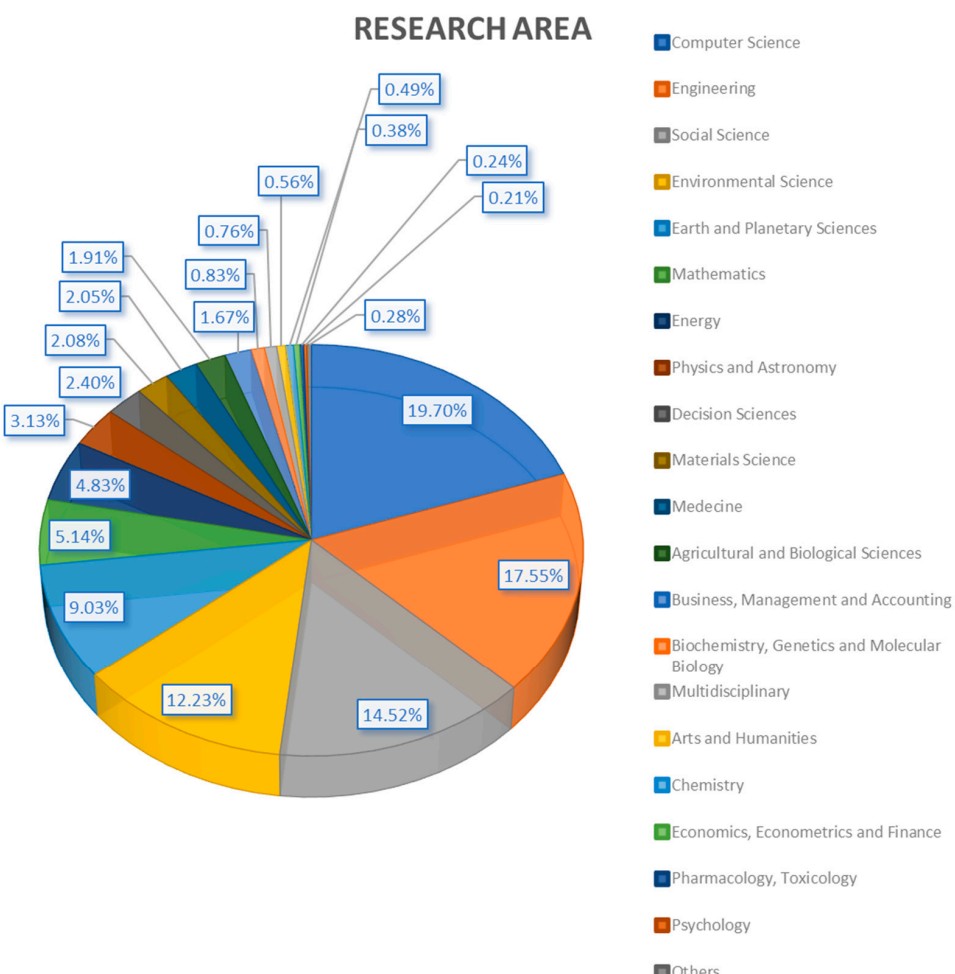

**Figure 2.** Machine learning on research areas (database: Scopus, authors' elaboration).

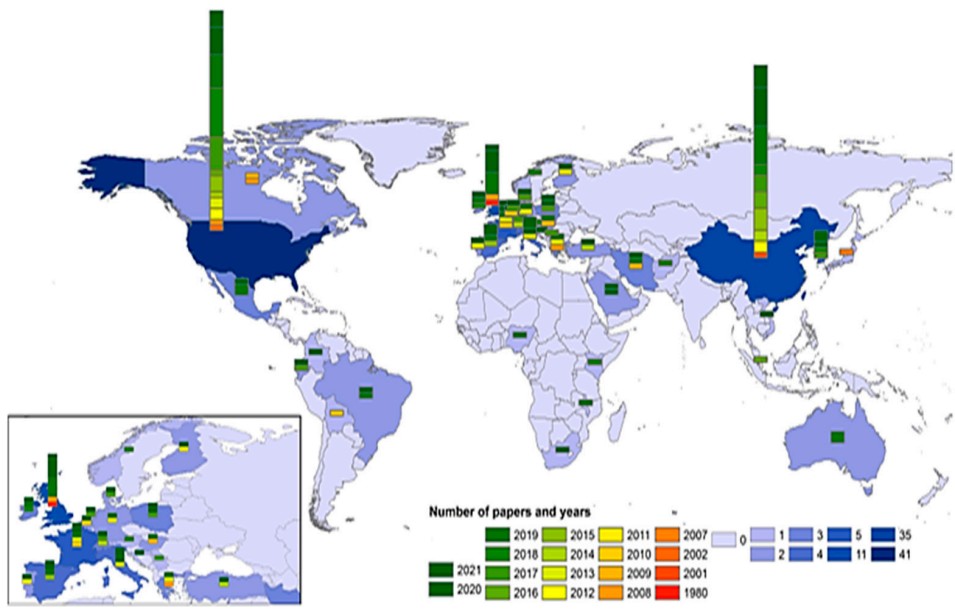

**Figure 3.** Spatial and temporal distribution of papers on ML related to urban applications by country and year [18].

### 3. Taxonomy of ML Methods for Urban Applications

Artificial Intelligence, globally, is divided into different parts, namely knowledge representation, genetic algorithms, Artificial Neural Networks (ANN), data mining, etc. The fields of urban planning and engineering are set to expand globally due to their strong, fast-growing relationship to data mining, especially in the smart cities' fields [21].

Despite the ML type and use, the quantity and type of data affect their accuracy and efficiency and enable (or not) the path towards the solution and alternative developments. In this process, Bhavsar et al. [2] underline the importance of problem definition for the appropriate application of ML methods (Figure 4). In reality, problem identification is a complex process depending on different factors, such as data mining, user skills and perceptions, etc.

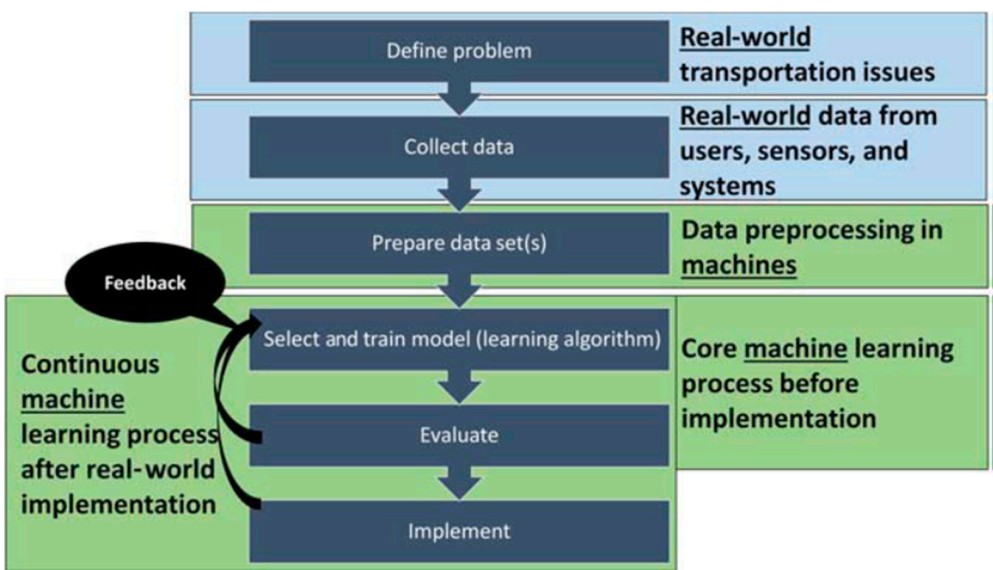

**Figure 4.** Machine learning algorithm steps.

Generally speaking, the ML methods are categorized based on the type of 'learning'. The most commonly known as follows [2]:

#### 3.1. Supervised Learning

Supervised learning methods deal with a function (or an algorithm) to compute outputs based on given information and present data (e.g., the number of dwellings per ha). This information will be used for an automated process to minimize the possible risks of a prediction error, expressed as the difference between the real (data) and the computed values. Examples of this ML are the binary classifications (True or False), etc., or regression problems.

#### 3.2. Unsupervised Learning

On the other side, unsupervised learning methods depend only on the unlabeled data and aim to identify hidden patterns of data. An example of this category is clustering, which focuses on the data grouping based on similarities or the method of association for the trends' identification concerning a specific problem.

#### 3.3. Machine Learning Algorithms: An Overview

However, the classification and taxonomy of ML require a thorough analysis of a set of attributes when discussing urban developments. Although there are many areas of focus, ML use has a major driver on land use and cover as great support for sustainable development. Nonetheless, despite the rise of smart cities and related concepts and the

advancement of big data, etc., there is little evidence regarding classification, simulation or predictions [22]; this section is an step towards the development of this ground.

Murphy [23] proposes three main types of ML methods, namely supervised (predictive) learning to identify a mapping from outputs to inputs considering a specific set of input-outputs, unsupervised learning, where only the inputs are given, and reinforcement learning, which is less commonly used and explain how to perform with the occurrence of given occasional rewards (Figure 5).

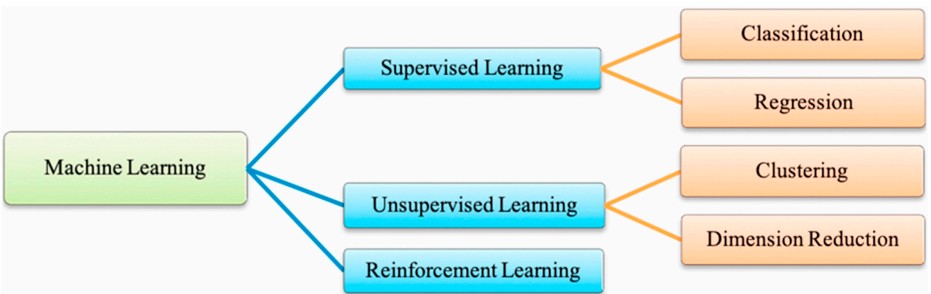

**Figure 5.** Taxonomy of ML common practices [23].

Emerging methods, such as convolutional neural networks (CNN), proved their efficiency in extracting features from spatial data [24], and recurrent neural networks (RNN) are promising approaches to accurate urban simulations. Examples of successful applications are found in various studies applied to road extraction from the wider perspective of both 2D optical remote sensing images and 3D point clouds commonly used for road data acquisition developed by Chen et al. [25,26]. In the same study, a comprehensive approach to the definition of morphological feature-based tools for road shape features is designed including support vector machines (SVM) ([27]). In the same study, Chen et al. provided three classifications for road area extraction based on traditional methods for identifying of road features (e.g., Lu et al. [28], Perciano et al. [29], etc.) or deep learning [30]).

Ensemble-based methods, such as random forests (RF) and similar methods, are boosted for the problem–solution studies of smart urban forms (e.g., [31–34]). On the other hand, ML methods are commonly used as a promising area to achieve smarter and more inclusive urban configurations in the tissues of modern cities [35].

*3.4. Decision-Making Urban Planning Processes*

Decision-making processes are fundamental in urban planning strategies, consisting, as they do, of simplified approaches to reality to enable decisions and interactions and allow planners to adjust or modify them in vitro using parametric proposals. Decision-support systems (DSS) facilitate the integration of models and enable the interactions between the diverse parameters to adjust or test solutions and evaluate the consequences leading to desirable and viable solutions. For the special case of predictive modeling, ML has been used for the identification of urban patterns and related indicators. Taking a quick look at the existing literature and the Scopus database correlations, one identifies of 585 documents for ML and the decision-making processes published in the United States and China, as presented in Figure 6.

Among the decision-making tree algorithms, the CART (Classification and Regression Tree) and ID3 (Iterative Dichotomizer 3) are the most commonly used for the land-use classifications acting as a random subset of the predicting parameters. On the other side, random forest (RF) has the particular functionality of both classification and regression analyses and handles an important volume of indicators [36]. In ML use, the DSS processes are usually represented in modeling for predictive forms to enable the decision and improve the design of the forms (Figure 7) [35].

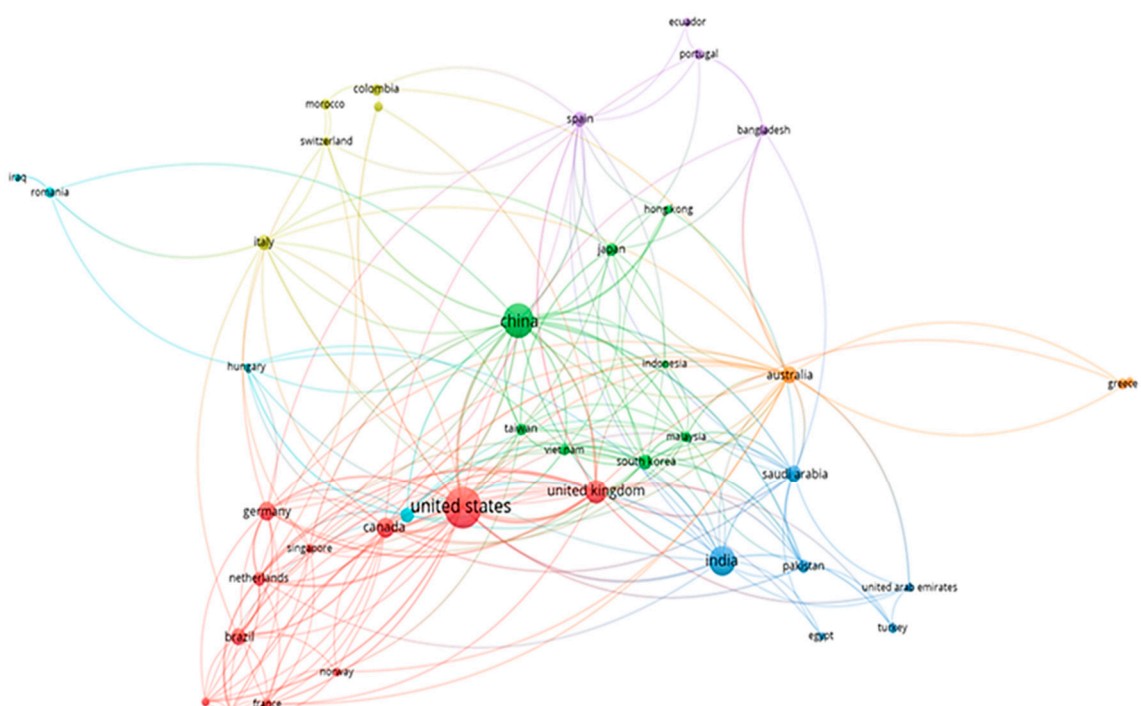

**Figure 6.** Correlations of ML and secision-making processes, Scopus database.

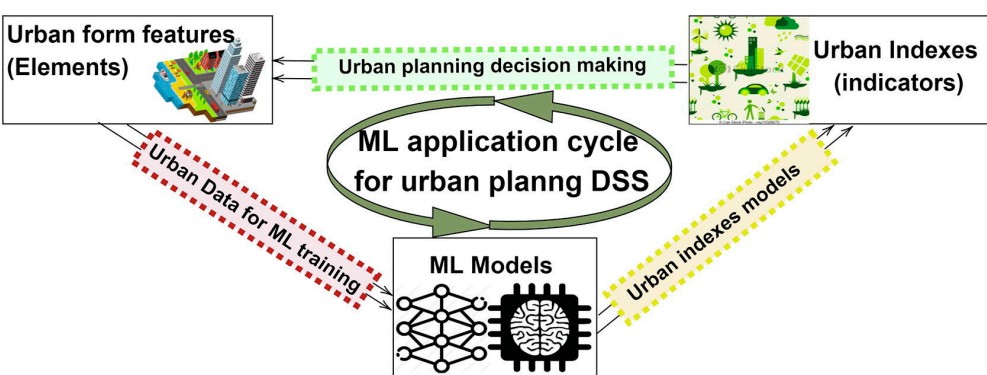

**Figure 7.** A cycle of ML application for urban form decision support system [35].

## 4. Review on Urban Applications

Within the wide access and implication of 'big data' and the increasing adoption of urban studies, numerous comprehensive insights are appearing in the literature for studying their potential for shaping the future urban environments in different sectors. The issue is the core of modern planning strategies aiming at digitalization and sustainable challenges. Specifically, to overcome the challenge of land use and cover, planners are integrating an ever-increasing complexity to qualify the dynamics and to manage the complexity [35].

Several ML algorithms have been tested for their performance on different forms of databases respecting the land-use management and/or simulation of land-use planning processes, with the more popular ones, supporting vector machines, neural networks, Markov random or GANS, experimenting on different datasets individually and in combination [37]. At the time being, there is a rising interest in ML use and applications.

Wu and Silva [38] reviewed the AI-based approaches in the projections of land use and their dynamics for spatial planning, Abdulijabbar et al. [39] related them to the mobility problems-solutions, while Yigitcanlar et al. [40] tackle the theme of 'sustainability'. Nevertheless, AI-based applications in urban contexts remain limited to supporting city

planning due to the dynamic systems the urban settlements present and the increasing amount of big data mining to obtain new knowledge. AI-based tools move from static to dynamic flows to forecast urban growth and enable spatiotemporal modelling [41] with the typical example being the agent-based modelling (ABM) method for the simulation of bottom-up processes to predict future city development. Patel et al. [42] recommended the GIS integration agent-based modelling for testing informal settings' policies in real-life urban environments, while Patt [43] investigated their applications with a focus on the public space networks.

To identify some representative scientific paradigms and trends, Table 1 summarizes the AI-based approaches in urban fields and explores the linkages of different factors, while the following sections focus on particular applications.

**Table 1.** Examples of AI-based approaches in city planning.

| Urban Theme | Scope | AI-Based Tools | Reference(s) |
|---|---|---|---|
| Polycentricity | Flow analysis and linkages, spatial simulations | Artificial neural networks, fuzzy logic, agent-based models | e.g., [31,32] |
| Spatial structures and dynamic analyses | Study on the functional structures of the city, mobility configurations, land-use identification | Artificial neural networks, fuzzy logic, agent-based models | e.g., [33,34] |
| Flows' analyses | Analysis of different types of flows in cities (e.g., energy, mobility, etc.) | Stochastic simulation models, artificial neural networks | e.g., [44–50] |
| Typo-morphological analysis | Analysis of urban structure, form and space | Stochastic models, Artificial neural metworks | e.g., [51–53] |

### 4.1. Machine Learning and Built Environment

Using ML to improve data collection and management is the priority of their users towards the solutions to complex problems, such as urban themes. Their power is incorporated for building energy efficiency, allowing the analytics of its operations and the identification of solutions to issues of performance and systems' behavior to reduce the use and improve the overall energy management. A typical representation of the ML use in buildings' optimization techniques is found in Kwok and Lee's work [54], as is the use of artificial neural networks (ANN) regarding the cooling predictions and increasing the level of accuracy with the use of fixed schedules and historical data related to the occupancy. Discussing the correlation of ML and energy, the building energy simulation (BEM) has a major role in low energy configurations and the development of advanced skills and knowledge towards clean energy [55].

Schoenfeld [56] cites the optimization in:

- Forecasting energy consumption to reveal trends and predict future energy uses and assist energy planning, management or conservation to reduce the energy demand and the $CO_2$ emissions [57] and alternative evaluations for an optimized operation to improve demand and supply balances [58]. Nonetheless, the demand for data collection (especially via intelligent sensors/meters) is evident. To that point, Ahmad et al. [59] underline the evolution of energy metering in technological terms, while Chammas et al. [60] analyze the importance of wireless networks and IoT-based methods for energy monitoring and relevant solutions. The literature unveils a significant number of studies related to forecasting activities and prediction performances; for example, Zhaond Liu [61] with respect to a highly-accurate prediction model for the building energy load with dynamic simulations;
- Detection and prediction of faults, in which traditional models do not provide preemptive interventions;
- Seasonality modeling, i.e., correlating themes to seasonal conditions.

The interest in ML use in buildings operations is rising and the AI-based solutions for the conception (design) and operation are found in numerous scientific works (e.g., [62–65]). On the other side, AI drives smart development and enables the advanced use of technologies for buildings' operation and maintenance, examples are found in the HVAC systems

or the lighting (e.g., [29,30]). Much research is being explored in AI use and integrated systems on the themes of energy efficiency, thermal comfort, etc. Tien et al. [55] visualize the transition from traditionalal to intelligent techniques (Figure 8).

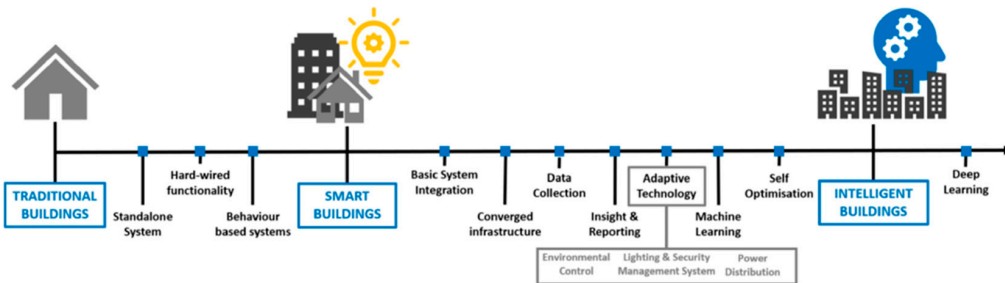

**Figure 8.** Evolution from traditional to smart techniques on buildings [55].

In line with construction and design, building information modeling—commonly acknowledged as BIM—is becoming the architectural norm to promote innovative and optimized designs with automated processes [66]. Nonetheless, this transition took approximately 25 years to be integrated into the market [67]; Gholizadeh et al. [68] explored the difficulties on this path, explaining that, as late as 2017, only three up to fourteen BIM functions were widespread in practical applications. The methodologies proposed are promising and incorporate both the design/conception and the construction processes for information delivery as a backbone to transcend organizational boundaries and mediate the gaps. The progress is notable considering that, in the absence of BIM and automated processes, scholars proposed stand-alone systems to represent the buildings; examples include HI-RISE for preliminary structural designs of tall buildings [69], SPEX for sizing structural cross-sections [70] or EIDOCC for the design of reinforced concrete [71]. Later on, natural language processing (NLP) was applied in design codes and regulations without further commercialization (e.g., [45,46]).

BIM enabled the automated project performance monitoring and control systems within the introduction of new concepts, e.g., 'Construction 4.0' or Digital Twins with diverse challenges from technical or conceptual standpoints, such as the integration of process information for comparison with monitored data and the need for sophisticated and complex approaches to its management (e.g., [72,73]) or ineffective production for planning and control systems [74].

Complementary to the cited approaches, Liu et al. [75] introduce the three-dimensional approach by three different methods: original building plans, field surveys and remote sensing technologies for the 3D interpretation of architectural features and extrapolation of relevant data and information, such as aerial images [76], light detection and ranging (LIDAR) data (e.g., [77,78]), satellite imagery [79] or even grouping-based stereo [80] or mono images using shadows or digital surface models [81].

### 4.2. Machine Learning and Land Use

Urban planning tools employ land use and cover to provide historical insights as a base for future urban development [82]. The land-use analysis uses remote sensing geographical information systems (GIS) to simulate the changes and reach a strategic decision for the designated area ([68,69]). Spatiotemporal land-use simulations as reproducible approaches for estimations and future land transitions are driving forces to support land-use policy decisions ([83–85]). These are relevant methods to this topic, commonly acknowledged with rising scientific research (e.g., [86–89]) and used is the cellular automata (CA) for the generation of urban patterns and nonlinear stochastic processes [90] and complicated interpretations of the complexity of the bottom-up model leading to the ignorance of land-use demand estimations [91].

On the other side, Machine Learning is becoming an imperative methodology to monitor ([92–94]) and forecast ([95–97]) the land-use challenges in urban areas. Undoubtedly,

statistical and spatial analyses proved their popularity (e.g., [98,99]). For many years, land-use mapping and modelling of geographical, demographical and relevant data have implicated ML as a vital tool to compose models to recognize urban configurations and minimize prediction errors utilizing learning strategies and related drivers [100].

In reality, data-driven models by ML have recently been recognized as powerful means for parametric approaches to land-use distribution (e.g., [101,102]) with multiple benefits, especially in dealing with massive amounts of information and numerous variables. The ability to model sophisticated and non-linear problems [103] used to be dependent on the SVM and the random forest (RF) [104].

Land administration is a core topic in urban planning strategies, which requires multi-faced information on built and non-built-up areas, functionalities, typologies and green and public spaces, to cite some of them. Dealing with this data is a sophisticated process, which includes open-source data provided by various repositories, commercially available satellite images, aerial photographs, cadastral boundary extractions, 3D modelling, etc. Nonetheless, the first stage of the territorial analysis usually demands demographic and statistical analysis of a given population with the need for a comprehensive geospatial database to assess the existing land use and estimate the future projections and the potential possible changes. Depending on the research scope, Chaturvedi and De Vries [11] provide a relevant classification in Table 2 [11].

**Table 2.** Land use planning indicators with measurements, data required and applications (adapted by [11]).

| Theme | Indicators | Data | Application |
|---|---|---|---|
| Urban expansion | Density, demographic profile, built/non-built | EO-based data (e.g., classified images, building footprints) | Classification and simulation (CNN, etc.) |
| Land restrictions | Land-use/cover, built/non-built-up spaces | A master plan, land-use regulations | Classification, and extraction of EO products (e.g., DEM), spatial logistic regression, Cellular automata |
| Land distribution | Policies, demographics | Census, socioeconomic data | |
| Zoning | Land-use distribution | Master Plan, classified images | Planned development |
| Land-use changes | Settlement patterns, urban growth processes, population growth | Spatiotemporal EO data | Spatial metrics, agent-based modelling |

Despite the support of evidence-based algorithmic processes, few provide supportive studies on the land-use theme; examples are found in Shafizadeh-Moghadam et al. [105] of the benchmarking land-use probability models. Karimi et al. [106] detailed the use of ML in land-use changes since 2011 in cellular automata, regression models, artificial neural networks, agent-based models, true-based models and support vector machines (SVM). ML also gained wide acceptance in transportation systems but with significant limitations due to its dependency on agents' predictions. Table 3 overviews the ML models on land use based on this study.

**Table 3.** Machine Learning use in land-uses strategies (adapted by Karimi et al. [106]).

| Machine Learning Use | Scope | Reference(s) |
|---|---|---|
| Cellular automata model (CA) | Land-use analysis related to transport and mobility systems (e.g., roads, railways, etc.) and population density issues | e.g., [107,108] |
| Artificial neural networks (ANN) | Annual population growth and land-use typologies | e.g., [83,84] |
| Linear regression models | Population density, land-use typology, economic centers analysis | e.g., [85,86] |
| Agent-based models (ABM) | Accessibility to functions and city infrastructure | e.g., [87,88] |
| Decision tree model (DT) | Land typologies, proximities to amenities, densities of residential, commercial | e.g., [109–115] |
| Support vector machines (SVM) | Land-uses typologies, built and unbuilt areas | e.g., [116–120] |

Nonetheless, an important challenge on this subject is the artificial absence of data handled by classification models with the commonly acknowledged MAXENT model (maximum entropy) as an advanced ML method originated from information theory ([121,122]). The cutting-edge method that the model proposes evaluates the spatial distribution and was first adopted for the early monitoring of illegal land development in line with the probability analysis and scenario development [123] based on the maximum entropy principles mainly used for the estimation of sustainable natural habitats and the occurrence probability of species, employing one-class classification of remote sensing imageries [124]. Overall, the maximum entropy algorithms minimize the amount of information and are used in ecological modelling (e.g., [121–125]).

- An overview of the correlations of the existing works and studies of ML and urban fields to identify the authors' names per year (1.372 documents) (Figure 9).

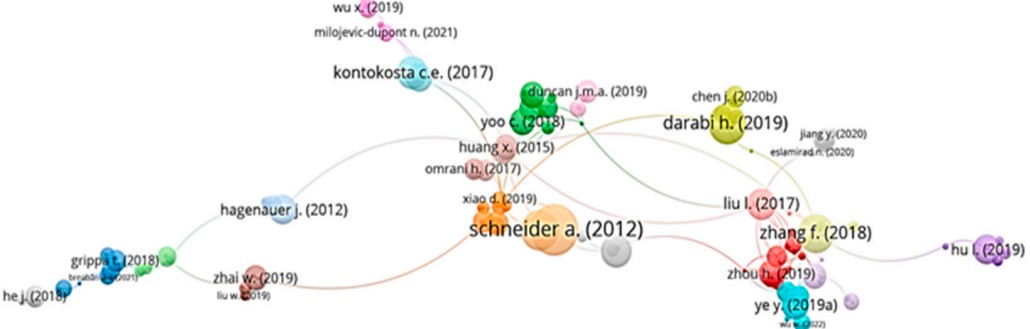

**Figure 9.** Bibliometric analysis of ML and urban applications (authors per year), Scopus database.

- An overview of the correlations of the existing works and studies of ML and urban fields to identify the number of citations per country (1.372 documents) (Figure 10).

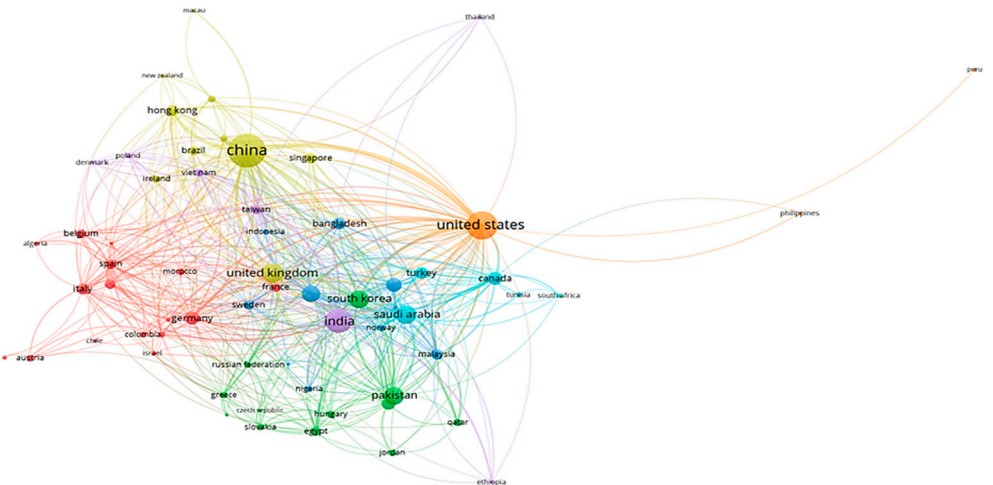

**Figure 10.** Bibliometric analysis of ML and urban applications (citations per country), Scopus database.

## 5. Examples of Case-Studies

Two typical examples of the existing case-studies portfolio of urban applications and ML implications are provided in this section, proving the importance of ML solutions to urban planning problems.

### 5.1. Shanghai Urban Drainage Masterplanning

Dealing with the challenges of rapid urbanization and urban growth (15 million people, a number that has tripled since 1990) to the impermeable area of the territory, Shanghai faces the threat of green space reduction and, consequently, of the rise of stormwater runoff. Considering the previous experiences of urban flooding in recent years, the city proposes

in its master plan for a horizon of 2035 an upgraded drainage system. Drawing upon its expertise and successful practices, ARUP, in a collaboration with the Shanghai Urban Construction Design and Research Institute, tailored a strategy to review the traditional approaches of drainage in the city being concentrated on the sensitive and integrated urban design and decentralized infrastructure by applying remote sensing imagery and ML technologies after a comprehensive territorial analysis of the elements concerning the urban fabric of the city (Figure 11) [126].

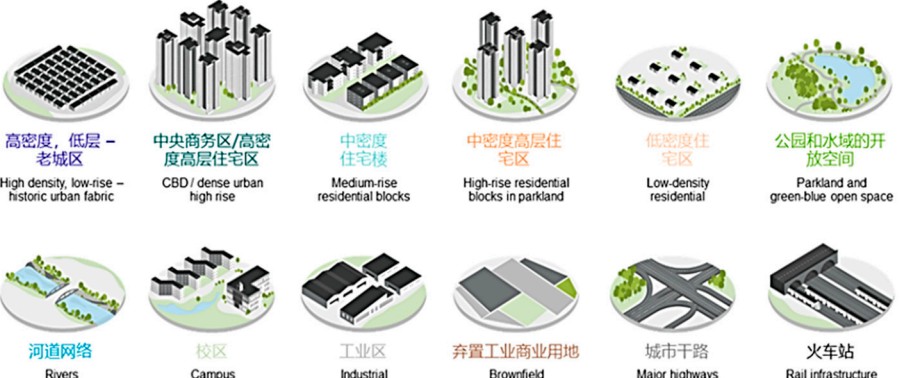

**Figure 11.** Land characterization of urban typologies of Shanghai.

In 2017, the design of the drainage masterplan had the specific objective of providing climate change adaptability and flood management strategies for the highly populated city center with three core objectives: reducing flooding, restoring clean water and delivering solutions to enhance the QoL in the city and the wider environment. The strategy meets the masterplan ambitions based on the following principles [127]:

- **Integrated:** considering the existing strategies in line with the 'Sponge City' and four elements: (a) a critical overarching system of governance mechanisms for collaboration and synergies; (b) green spaces to promote nature-based solutions (NBS); (c) blue equipment for flood defenses and relevant infrastructure; and (d) 'grey' equipment for drainage treatment (e.g., pumps, etc.);
- **Adaptive:** development of flexible approaches for risk management and uncertainties;
- **Smart:** integration of intelligent and digitalized models for optimization and data treatment of sophisticated scenarios based on planning strategies alongside the stormwater conditions.

Within the ML use, the concept was based on machine learning, artificial intelligence and open-source observations to identify urban typologies and NBS to address the diverse challenges proposing integrated models to improve stormwater management. Based on this approach, remote sensing tools were used for the city scanning and the classification of 12 categories required flooding protection to provide the necessary approaches for targeted water management, which facilitated the decision-making processes around the nature-based solutions beyond economics (Figure 12).

*5.2. MassMotion Pedestrian Simulation*

Another interesting application developed by ARUP (and commercialized through Oasys, 2014) is the MassMotion simulation for pedestrian movements in a city within ML use and the BIM tools use in a human-oriented design and approach. The goal of this application is to simulate people's movements and optimize the users' journeys in the city in an 'intelligent' way using real-time data, developing at the same time the ability to develop and test multiple alternatives and scenarios in graphical structures and allowing optimal decision in city planning.

Along with a 3D interface, the tool replicates the city models considering a series of actors (agents) and their interactions with the possible planning strategy and evaluates their roles in diverse simulation scenarios. Each of these agents has a specific origin and

destination and an outset of microsimulations is performed with the ability to consider alternative solutions adapted to the specific problem [128].

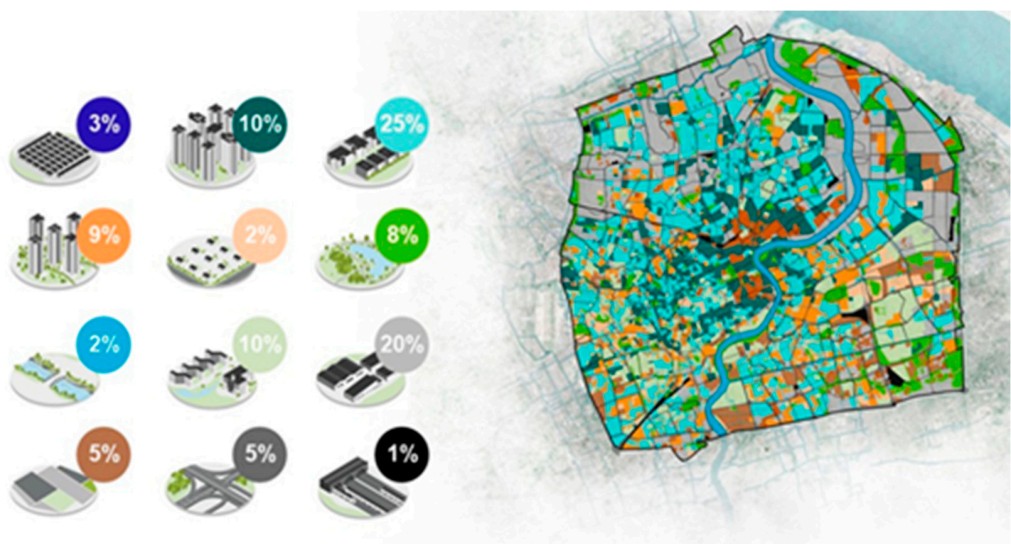

**Figure 12.** Machine Learning use the master planning area in different urban configurations.

Montjoy [129] characterized MassMotion as one of the world's most advanced simulation tools at the conception phase with strong visualization capabilities. As a stand-alone software, MassMotion illustrates the flows and densities in peak hours, considering crucial factors such as the speed, direction, etc., and creates geometrical models and designs for the planning of the studied transportation hubs; for example, those of the new Trondheim Central station (Norway) project; it defines of future conditions by developing optimized adjustments and validates the possible scenarios to obtain a well-functioning and human-centered station. Examples of this action are the analysis of the pedestrian patterns for the detection of congestion hotspots or crowded spaces, visibility and safety issues and the vertical and horizontal connections. The project lays on the flexibility of the design choices for future expansions by using forecasted passenger data of the related functions in complex 3D environments (Figure 13).

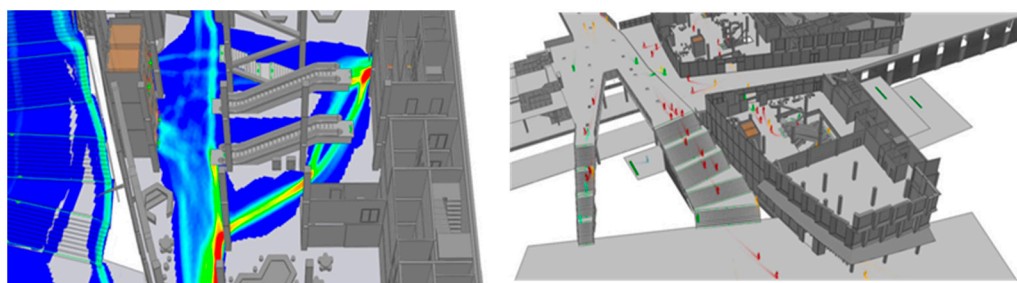

**Figure 13.** MassMotion applications in 3D complex environments.

## 6. Conclusions

As living laboratories, cities are facing tremendous complexities in accommodating a growing population and meeting the challenges of climate change and equal living for their citizens. Urban efficiency is being leveraged by uncertain environments with increasing transformations.

Being at the intersections of computer science, statistics and informatics, ML methods lead to more evidence-based solutions and decision-making processes along with the proposal of the dynamics of urban drives. They meet the challenge of 'big data' with a huge potential for smart and sustainable planning leading to resilient and inclusive urban

configurations. From these perspectives, ML addresses socioeconomic issues including the challenges of inclusiveness, poverty, and environmental and sustainability issues. The smart city indicators related to digital integration and data management shaping a city more intelligently and autonomously could estimate and evaluate its evolutionary trends (e.g., land-use evolution and definition of estimated needs), anticipate phenomena (or crises) and regulate them accordingly to better direct the growth in a long-term horizon. ML provide new opportunities to better monitor, understand and predict the future and guarantee the wellbeing of future generations

In this digitalized era, and with the rapid growth of computational skills and advancements in artificial intelligence, ML uses in various applications are gaining a rising interest from scholars and practitioners and gaining popularity in many research fields. A particular lever of their implications is being developed in the framework of smart city development and urban design with the use of geospatial data in different aspects of the urban system consisting of multiple tangible (e.g., land use and coverage) and intangible aspects (e.g., social inequalities).

Reflecting the increasing interests in ML uses, several approaches have been proposed in the existing literature towards the direction of enhancing urban dynamics that go beyond the traditional techniques of urban modelling, which is an indispensable tool for planning decision support. A remarkable potential for addressing urban challenges is found in ML methods (e.g., land use/cover, energy efficiency, etc.) consisting of spatiotemporal analyses.

The key contribution of this paper has been to provide a critical angle on the ML taxonomy with respect to its use of the urban planning sector, the methods and tools for urban problems and associated challenges and future research directions. It discusses two representative examples of city applications developed by ARUP. In a scoping review, the authors discussed the insights from an urban planning view to identify the gap in specific applications concerning built and urban environments. They also provided an overview of the existing aspects of the field along with systematic reviews and a thorough bibliometric analysis through a database search (Scopus) to ensure the highest academic standards and the validity of the relevant outcomes, screening and review conduct in charting the main components of the topic.

The ML integration into urban planning strategies will meet the evolutionary challenges in its analysis, simulation and monitoring of the future urban form and its applications. However, there is still an uneven distribution in this area with limited studies that address the challenges and gains of its use for future urban developments. New methods are needed to link the research on ML and urban science with the use of big data and evidence-driven shifts in order to connect these analytic frameworks and support further synergies and in-depth explorations of the practical issues of city challenges, and to increase reproducibility with the construction of a common language and protocols.

**Author Contributions:** Conceptualization S.K. and C.S.I.; methodology, S.K.; investigation, S.K.; writing—original draft preparation, S.K.; writing—review and editing, S.K. and C.S.I.; supervision, C.S.I.; project administration, C.S.I. All authors have read and agreed to the published version of the manuscript.

**Funding:** This research received no external funding.

**Institutional Review Board Statement:** Not applicable.

**Informed Consent Statement:** Not applicable.

**Data Availability Statement:** Not applicable.

**Conflicts of Interest:** The authors declare no conflict of interest.

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
