# Peer review of "Unveiling the Potential of Machine Learning Applications in Urban Planning Challenges"

_land, doi:10.3390/land12010083_

Round 1

Reviewer 1 Report

The title of the paper is new and interesting, but still, there are some flaws that need to be addressed by the author before publication.

1.       The abstract of this paper needs to be revised in terms of its applications and contributions.

2. The introduction and the proposed methodology are well described. However, the authors are suggested to state the motivation and objective of this paper more clearly in the manuscript.

3. The related work section is comprehensive, but there are some inconsistencies and problems in the background and related work section.

4. Add a separate paragraph in the introduction which shows the novelty of the proposed work.

5.     A few of the references are missing some information; you may complete them critically.

Author Response

Dear reviewer,

thank you for your valuable feedback, time and consideration to our paper. The authors read carefully your remarks and suggestions and provided the relevant modifications (see track changes of the updated version of the manuscript).

Please find enclosed the report with the detailed analysis for your queries.

Sincerely,

The authors team

Reviewer 2 Report

This manuscript (land-2103448) makes an interesting literature review because: (1) the large number of manuscripts that are considered and (2) the relevance of the topic to this distinguished journal: machine learning and urban land use planning. The link between scientific research and practice is still missing and I therefore appreciate the authors’ efforts. That being said, several improvements should also be done to enhance its potential value. My main concern is that much more works need to be considered because a number of related research was still ignored. Detailed comments and suggestions are listed as follows:

(1) In the Introduction Section, the authors devote too much space (Line 22~Line 31) to describing the well-known background of urban machine learning, which I think could be briefly mentioned in several sentences.

(2) The differences between this review and some previous related reviews need to be compared and discussed. Although two or three of them have been mentioned in the current manuscript, detailed comparison and discussion are still missing. What is the progress of this review, as comparing with the previous related review articles?

Machine learning for spatial analyses in urban areas: a scoping review. Sustainable Cities and Society, 2022, 85:104050

(3) Define what is the meaning of "Qol".

(4) The Figure 1 is not that necessary. It is better to clarify the logic framework of the literature review, that is, the connection among the different subsections.

(5) The authors failed to clearly explain which database they have searched for (only Scopus?). And more importantly, what were the searching keywords? This information is very important for reviewing this manuscript.

(6) Figure 3. Bibliometric Analysis of ML and Urban Applications (authors per year): this figure did not convey any meaningful information because only the authors' last names have been mentioned. What information can we infer from this figure? The same question for the Figure 7.

(7) In Section 2.3 Machine Learning Algorithms an overview, I suggest that a very unique type of one-class machine learning techniques, maximum entropy (MAXENT) model, should also be discussed in this section (see below for example). The principle of maximum entropy is a model creation rule that requires selecting the most unpredictable (maximum entropy) prior assumption if only a single parameter is known about a probability distribution.

Estimating potential illegal land development in conservation areas based on a presence-only model. Journal of Environmental Management, 2022, 321:115994

Land subsidence hazard modeling: Machine learning to identify predictors and the role of human activities. Journal of Environmental Management, 2019, 236:466-480.

(8) I wonder whether Conference Paper, Conference Review, and Letter were selected for literature review? The authors did not explain this important information in the current manuscript.

(9) The connection and the logic among all these different sections should be clear stated at the beginning of this manuscript. I think a flowchart may be helpful for explanations. It should provide an overview of the key points of each source and combine them coherently. In the current manuscript, it is suggested to discuss the significance of findings in relation to the literature as a whole.

(10) In addition, it is better to summarize the key findings and emphasize their significance in the Discussion and Conclusion Sections. Also, discuss how this review has developed the research framework by using the theories and methods.

(11) In Section 3.1 Machine Learning and Built Environment, I suggest that the application of machine learning on three-dimensional built environment should also be discussed in this section (see below for example).

Landscape metrics for three-dimensional urban building pattern recognition. Applied Geography, 2017.

(12) In Section 3.2 Machine Learning and Land-use, I suggest that the latest cellular automata model, i.e., the FLUS model, should also be discussed in this section (see below for example).

(13) The whole Conclusion Section is a bit general and theoretic. This part could be improved by found it on a more scientific and quantitative base. What is new and what do we learn from this? Specify more what are the issues and the advantages of the different types.

Author Response

(The authors gave the same response as above.)

Reviewer 3 Report

Interesting subject but needs probably some further expansion/analysis and discussion from section 3.2 and onwards

Please review the text and correct in several places the syntax of english language

Author Response

(The authors gave the same response as above.)

Round 2

Reviewer 1 Report

Author has incorporated all the changes , now paper is ready for acceptance.

Reviewer 2 Report

I appreciate the authors' efforts to improve this manuscript. Now it is acceptable for publication.

Reviewer 3 Report

No comment